# Storage Compound Accumulation in Diatoms as Response to Elevated CO_2_ Concentration

**DOI:** 10.3390/biology9010005

**Published:** 2019-12-24

**Authors:** Erik L. Jensen, Karen Yangüez, Frédéric Carrière, Brigitte Gontero

**Affiliations:** CNRS, BIP, UMR 7281, IMM, FR3479, 31 Chemin J. Aiguier, Aix Marseille Univ., 13 402 Marseille CEDEX 20, France; erikjensenrojas@gmail.com (E.L.J.); kyanguez@gmail.com (K.Y.); carriere@imm.cnrs.fr (F.C.)

**Keywords:** CO_2_, chrysolaminarin, diatoms, triacylglycerol

## Abstract

Accumulation of reserve compounds (i.e., lipids and chrysolaminarin) in diatoms depends on the environmental conditions, and is often triggered by stress conditions, such as nutrient limitation. Manipulation of CO_2_ supply can also be used to improve both lipids and carbohydrates accumulation. Given the high diversity among diatoms, we studied the two marine model diatoms—*Thalassiosira pseudonana* and *Phaeodactylum tricornutum*, a freshwater diatom, *Asterionella formosa*, and *Navicula pelliculosa*—found in fresh- and sea-water environments. We measured the accumulation of reserve compounds and the activity of enzymes involved in carbon metabolism in these diatoms grown at high and atmospheric CO_2_. We observed that biomass and lipid accumulation in cells grown at high CO_2_ differ among the diatoms. Lipid accumulation increased only in *P. tricornutum* and *N. pelliculosa* grown in seawater in response to elevated CO_2_. Moreover, accumulation of lipids was also accompanied by an increased activity of the enzymes tested. However, lipid accumulation and enzyme activity decreased in *N. pelliculosa* cultured in fresh water. Chrysolaminarin accumulation was also affected by CO_2_ concentration; however, there was no clear relation with lipids accumulation. Our results are relevant to understand better the ecological role of the environment in the diatom adaptation to CO_2_ and the mechanisms underpinning the production of storage compounds considering diatom diversity.

## 1. Introduction

Diatoms are important primary producers in fresh and seawater environments. They contribute up to 20% of global CO_2_ fixation [1]. Carbon partition in diatoms involves the production of carbohydrates, specifically the soluble carbohydrate chrysolaminarin, and lipids (as triacylglycerols; TAGs) as main storage compounds [2]. Chrysolaminarin is stored in the vacuole [3], while lipids are accumulated in diatoms mainly as oil bodies in the cytoplasm and, in less amounts, in the chloroplasts [4].

Lipid production in microalgae has gained increasingly attention for their potential in industrial applications, including the production of third generation biofuels [5,6,7] and pharmaceutical and cosmetical products [8]. Lipid accumulation in some microalgae species can reach up to 73% of their dry weight and show higher productivities compared to crop plants [9,10]. Within microalgae species, diatoms have emerged as a good source of lipids because of their exceptional photosynthetic efficiency and their ability to accumulate high amounts of lipids in many different culture conditions [6,11,12,13].

Several studies show that lipids can accumulate in diatoms after mild or acute nutrient deprivation, generally at the expense of a decrease in biomass [14,15]. Increase in lipid accumulation after nitrogen starvation has been observed in some diatom species, such as *Phaeodactylum tricornutum*, *Chaetoceros muelleri*, and *Thalassiosira weissflogii* [16,17]. Other nutrient limitation stresses involved in an increased lipid accumulation are phosphorous and silicon deprivation [17,18]. In *Chaetoceros gracilis*, a combined mild deprivation of both silicon and sodium chloride produced a lipid accumulation reaching up to 73% of the total cell dry weight [10]. Other approaches to enhance lipid production in diatoms involve the use of phytohormones and other hormone-derived compounds [19,20], as well as the pharmacological modulation of signaling pathways involved in the synthesis of lipids [20,21]. In addition, thanks to the new molecular engineering approaches in diatoms, it is also possible to enhance lipid accumulation in these organisms by genetic transformation. For instance, overexpression of several genes involved in the biosynthesis of TAGs can increase lipid accumulation in *P. tricornutum* (reviewed in [18]). It can be expected that metabolic engineering will be more widely used as more sequences of diatom genomes become available.

Chrysolaminarin has been less studied compared to lipids, however there is a growing interest in its production because of its potential biotechnological applications, including its antioxidant and immunomodulatory effects in aquaculture, as well as its antitumor effects in human cancer [22,23,24]. In the diatom *Skeletonema costatum* under severe nutrient deprivation and subsaturating light intensity, it has been shown that chrysolaminarin can accumulate up to 80% [25].

CO_2_ is essential for diatom growth and metabolism and since the synthesis of storage compounds requires a carbon supply, it is likely that increased carbon availability might also improve lipid accumulation. In agreement with this hypothesis, some studies have shown that diatoms accumulate more lipids under high CO_2_ conditions [26,27,28]. In fact, the mechanisms of carbon capture and assimilation have been studied for many years [29], for instance, the CO_2_-concentrating mechanisms (CCMs), which involve bicarbonate transporters and carbonic anhydrases whose expression is often modulated by environmental CO_2_ concentrations [30,31,32]. However, the effect of CO_2_ on biomass and lipid accumulation is yet poorly understood and is also overlooked on chrysolaminarin production. Furthermore, carbon fixation and metabolism is a tightly regulated process and diatom regulation of CO_2_ fixation (i.e., the Calvin–Benson–Bassham cycle) differs in many aspects from that of green algae and land plants [33]. In addition, to date, most studies regarding the effects of CO_2_ and the production of storage compounds in diatoms have been done on marine species, and much less is known about freshwater species.

Diversity makes diatoms attractive models to study their capacity to produce either lipids or carbohydrates and we therefore studied different species. In this work, we studied four diatom species, the two most explored diatoms living in seawater environment: the centric (*Thalassiosira pseudonana*) and the pennate (*Phaeodactylum tricornutum*) diatom; in parallel, we also studied the pennate freshwater diatom *Asterionella formosa* and a pennate species capable to grow both in seawater and in freshwater environments (*Navicula pelliculosa*). We analyzed the production of their main storage compounds at two different CO_2_ concentrations (400 and 20,000 ppm).

## 2. Materials and Methods

### 2.1. Strains and Culture Conditions

*Thalassiosira pseudonana* Hasle & Heim., strain CCMP1335 subcultured in the lab since 2013, *Phaeodactylum tricornutum* Bohlin, strain CCMP2561, and *Navicula pelliculosa* (Brébisson ex Kützing) Hilse, strain CCAP 1050/9, were grown in artificial sea water enriched with Guillard’s “F/2” nutrients plus silicon (F/2+Si) as described previously [34]. The diatom *N. pelliculosa* was also cultured in a freshwater medium in the absence of NaCl and KCl, and NaH_2_PO_4_·2H_2_O (0.036 mM) was replaced by KH_2_PO_4_ (0.036 mM). *Asterionella formosa* Hassal, isolated from the English Lake District (Esthwaite Water; strain BG1), was grown in Diatom Medium (DM) as described [19]. Cultures were maintained at 18 °C in a growth cabinet (Innova 4230; New Brunswick Scientific, Edison, NJ, USA) with continuous light at 50 µmol photon m^−2^ s^−1^, constantly shaken at 90 rpm, and bubbled with air containing either 400 ppm or 20,000 ppm CO_2_ at a gas flow rate of 320 mL min^−1^ L^−1^ of culture. For growth curves, optical density was followed at 750 nm using a Perkin Elmer Lambda 25 UV/VIS spectrophotometer (Waltham, MA, USA).

### 2.2. Biomass Determination

Twenty-five to 50 mL of cell culture were filtered using a 0.45 µm nylon filter membrane (Merck-Millipore, Burlington, MA, USA), then dried overnight in an oven at 60 °C. Cell dry biomass was determined by the difference in the weight of the dried membrane versus the same membrane before filtering cells. Biomass productivity was determined as in Hempel et al. [35], using the formula *P_biomass_* (mg L^−1^ day^−1^) = (*X*_2_ − *X*_1_)·(*t*_2_ − *t*_1_)^−1^, where *X*_2_ and *X*_1_ correspond to the biomass concentration (mg L^−1^) at the end (*t*_2_) and start (*t*_2_) time of cultivation, respectively.

### 2.3. Lipid Extraction and Measurement

Lipids were extracted as previously mentioned in [21]. Diatom cultures of 250 mL were harvested by centrifugation at 3500 g for 20 min. Pellets were immediately resuspended in 2–3 mL of a chloroform/methanol solution (2:1, v/v) plus 0.2 mL of 1 N HCl and kept at −20 °C in glass vials. To complete lipid extraction, samples were mixed with one volume of ultrapure H_2_O and vortexed vigorously. The mixtures were centrifuged at 1000× *g* for 5 min at 4 °C to fully separate the two phases. The lower (organic) phase was recovered and washed once with a NaCl-saturated solution, followed by centrifugation at 3500× *g* for 20 min and the organic phase was recovered again. The lipid-containing organic phase was finally transfer to a glass vial for storage (−20 °C). Neutral lipids were separated by thin layer chromatography on silica-coated rods (SIII Chromarods) and detected by flame ionization using a MK-6 Iatroscan TLC-FID apparatus (Iatron Laboratories, Tokyo, Japan) as described previously in [36]. Data acquisition and processing was performed with the i-Chromstar 6.3 integration software (SCPA GmbH, Bremen, Germany). The amount of TAGs was estimated from a calibration curve. Lipid productivity (mg TAGs L^−1^ day^−1^) was calculated as *P_biomass_*·*Cf*, where *P_biomass_* is the biomass productivity, calculated as shown above, and *Cf* corresponds to the final concentration of TAGs at exponential or stationary phase.

### 2.4. Chrysolaminarin Measurement

Chrysolaminarin was measured according to the method performed by Granum and Myklestad [37] with some modifications. Briefly, 2 mL of cell culture were centrifuged at 3500× *g* for 15 min at 4 °C. Cell pellet was resuspended in a solution of 50 mM H_2_SO_4_ and incubated 1 h at 60 °C. The suspension was then centrifuged at 16,000× *g* for 10 min at room temperature. The supernatant was recovered and put in an oven at 60 °C. Dried samples were resuspended in 1 mL of concentrated H_2_SO_4_ and 30 µL of a fresh solution of 3% phenol and incubated at 85 °C for 1 h. Absorbance was measured at 493 nm and chrysolaminarin content was calculated using a calibration curve with glucose as standard. Cells were counted prior to chrysolaminarin extraction and measurement, using a hemocytometer (improved Neubauer chamber).

### 2.5. Protein Extraction

Cultures of 250–500 mL were centrifuged at 3500× *g* for 10 min at 4 °C. Pellets were resuspended in buffer 20 mM TRIS, 50 mM NaCl (pH 8), plus lysozyme (final concentration: 40 µg mL^−1^) and a protease inhibitor cocktail (Sigma^®^, St. Louis, MO, USA; Concentrations: 2 mM AEBSF, 0.3 μM Aprotinin, 116 μM Bestatin, 14 μM E-64, 1 μM Leupeptin, and 1 mM EDTA). Cells were broken using an ultrasonicator (Sonics & Materials Inc, Vibracell, Bioblock, Danbury, CT, USA) with four cycles consisting of 10 pulses followed by 1 min pause (40% power, 60% pulse). Lysates were centrifuged at 16,000× *g* for 30 min at 4 °C and the supernatant was collected and kept at −80 °C for further analysis.

### 2.6. Enzyme Activity Measurements

Prior to glyceraldehyde 3-phosphate dehydrogenase (GAPDH) measurements, bisphosphoglyceric acid (BPGA) was synthesized. BPGA synthesis was performed as follows: a reaction containing phosphoglyceric acid (PGA; 66 mM), 33 mM ATP and ~4.5 units of phosphoglycerate kinase (PGK) was incubated in glycyl-glycine buffer (50 mM glycyl-glycine, 0.5 mM EDTA, 15 mM MgCl_2_, 50 mM KCl, pH 7.7) for 20 min at 30 °C. To determine GAPDH activity the consumption of NADPH to NADP^+^ (ε^340 nm^ = 6 220 M^−1^ cm^−1^) was followed using a Perkin Elmer Lambda 25 UV/VIS spectrophotometer (Waltham, MA, USA) at 340 nm in a reaction mix containing 50 µL of the freshly prepared BPGA, 0.2 mM NADPH, and 2–20 µL of protein crude extract in a total volume of 1 mL in buffer. Additionally, dithiothreitol (DTT) might be added at 5 mM final concentration for redox assays.

Phosphoglycerate kinase (PGK) activity was measured indirectly by coupling the PGK-mediating synthesis of BPGA to the NADH-consuming GAPDH-mediated synthesis of glyceraldehyde 3-phosphate (G3P). The reaction mix contained 1 mM ATP, 2 mM PGA, 0.15 mM NADH, and 0.5 unit of rabbit GAPDH in a total volume of 1 mL. PGK activity was related to NADH consumption followed at 340 nm as previously.

For pyruvate kinase (PK) activity, the conversion of phosphoenolpyruvate (PEP) to pyruvate catalyzed by PK was coupled to lactate production by the lactate dehydrogenase (LDH), which produces lactate using pyruvate as substrate, and consuming NADH. The reaction mix contained 5 mM PEP, 0.2 mM NADH, 2 units of LDH, and 1 mM ADP in a total volume of 1 mL, and NADH consumption was followed at 340 nm.

### 2.7. Statistical Analysis

Unpaired *t*-test analysis was performed to compare two group of samples using Graphpad Prism 6 software. Three independent biological replicates were done for each experiment, unless stated otherwise in the main text or figure legends, and comparisons done between groups of cells grown at high CO_2_ versus low CO_2_.

## 3. Results

### 3.1. Growth and Biomass as Response to High CO_2_

The growth of seawater and freshwater diatom species grown at atmospheric CO_2_ (400 ppm; low) or at high CO_2_ (20,000 ppm) was followed by measuring the optical density of the cultures at 750 nm (Figure 1).From all diatom species studied, the growth of *A. formosa* was the most affected in cells grown at high CO_2_ and the calculated growth rate was 0.57 ± 0.09 day^−1^ at high CO_2_ and 0.26 ± 0.02 day^−1^ at low CO_2_. In contrast, the growth rate of the other diatom species did not show significant differences (not shown) and their growth curves were only slightly or even not affected by CO_2_ (Figure 1). Nonetheless, we observed that the growth of *T. pseudonana* cells cultured at high CO_2_ was arrested after six days with a concomitant strong cell flocculation and precipitation, impeding the continuity of these cultures.

Biomass productivity was significantly increased only in *T. pseudonana* in the stationary phase and in *N. pelliculosa* grown in sea water in both exponential and stationary phase as response to high CO_2_ (Table 1).

### 3.2. Storage Compounds (TAGs)

As expected, TAG accumulation in the stationary phase was higher than in the exponential phase for most studied diatoms, except for the freshwater diatom, *A. formosa* (Figure 2e, Table 1). At high CO_2_ TAG productivity, in the stationary phase, for *P. tricornutum* and *N. pelliculosa* cultured in seawater medium significantly increased by 1.7-fold and 1.4-fold, respectively (Figure 2b,c), while production of TAGs was barely present in the exponential phase regardless of the CO_2_ treatment. In contrast, in the exponential phase, *N. pelliculosa* cultured in freshwater medium accumulated higher levels of TAG than in the seawater medium, with no significant difference between high and low CO_2_. On the other hand, in the stationary phase, *N. pelliculosa* grown under high CO_2_ had lower TAG content compared to low CO_2_ (Figure 2d), contrary to what was observed in *N. pelliculosa* grown in seawater medium (Figure 2c and Table 1). In *T. pseudonana,* TAG accumulation was not significantly affected by high or low CO_2_ neither at the exponential nor at the stationary phase (Figure 2a). TAG production in *A. formosa* was much lower than those of the other diatoms analyzed, and of interest, there was no TAG accumulation in the stationary phase of cells grown at low CO_2_ (Figure 2e).

### 3.3. Storage Compounds (Chrysolaminarin)

The contents of the carbohydrate chrysolaminarin in the diatom species studied were different and ranged from ~1 to up to 40 pg cell^−1^ (Figure 3). In *T. pseudonana*, the chrysolaminarin content from the cells grown at high CO_2_ compared to low CO_2_ significantly decreased by 0.38- and 0.44-fold in the exponential and stationary phase, respectively, (Figure 3a). In contrast, in *P. tricornutum,* the chrysolaminarin production was similar in the exponential phase of cells grown at low or high CO_2_; however, a significant 1.5-fold increase was found in the stationary phase of cells grown at high CO_2_ (Figure 3b). Only a 4.4- and 2.6-fold increase in the stationary phase, compared to the exponential phase, was observed in *N. pelliculosa* grown either in seawater or in freshwater medium, respectively, with no effect of CO_2_ concentration in neither of the two cultures (Figure 3c,d). *A. formosa* accumulated up to 40 pg of chrysolaminarin cell^−1^ at low CO_2_ and a strong 65% reduction was observed in cells grown at high CO_2_ versus low CO_2_ at the exponential phase (Figure 3e).

### 3.4. Activities of Key Enzymes from Carbon Metabolism

#### 3.4.1. Glyceraldehyde 3-Phosphate Dehydrogenase (GAPDH)

The NADPH dependent activity of the chloroplast glyceraldehyde 3-phosphate dehydrogenase (GAPDH) was measured from crude extracts of cells maintained at 400 ppm or 20,000 ppm CO_2_ (Figure 4).

In *T. pseudonana*, GAPDH activity and its regulation by metabolites were similar for cells grown at 400 or 20,000 ppm (Figure 4a). Whatever CO_2_ conditions during the growth, GAPDH was strongly activated by its cofactor, NADPH (0.2 mM) alone, but not by the reducing agent, dithiothreitol (DTT) alone (5 mM). However, for this diatom, GAPDH activity was slightly increased by pre-incubation with both NADPH and DTT compared to pre-incubation of these compounds alone.

In contrast, the activity of GAPDH from *P. tricornutum* cells grown at 400 ppm CO_2_, required, prior to assay, pre-incubation with both DTT and NADPH (Figure 4b). However, GAPDH extracted from cells grown at 20,000 ppm CO_2_, was always active whatever the pre-incubation mixtures and its activity increased up to 8-fold compared to GAPDH mixed with DTT and NADPH prior to assay and extracted from cells grown at 400 ppm. The GAPDH from *N. pelliculosa* cultured in seawater or freshwater media and grown at 400 ppm CO_2_ was only activated by reducing conditions (i.e., with DTT; Figure 4c,d). Like in *P. tricornutum*, the enzyme was active in all conditions tested for cells grown at 20,000 ppm CO_2_ in seawater and freshwater media. Moreover, the GAPDH activity value in *N. pelliculosa* was 2.3-fold greater when cells were cultured in the seawater medium compared to freshwater for cells grown at high CO_2_. GAPDH activity in *A. formosa* only slightly increased, but not significantly, when cells were grown at 20,000 ppm CO_2_ (Figure 4e) and, for cells grown at both CO_2_ concentrations, activity was only inhibited after treatment with NADPH in oxidized conditions (i.e., without DTT).

#### 3.4.2. Phosphoglycerate Kinase (PGK)

PGK activity was also measured from cell crude extracts. No changes were observed for PGK activity of *T. pseudonana* cells grown at 400 or 20,000 ppm CO_2_ (Figure 5a). PGK activity increased in *P. tricornutum* and *N. pelliculosa* (seawater) cells grown at 20,000 ppm CO_2_ by around 5- and 2.5-fold, respectively (Figure 5b,c). In contrast, PGK activity of *N. pelliculosa* cultured in freshwater medium and grown at 20,000 ppm CO_2_ decreased by 60% (Figure 5d). Moreover, PGK activity in *N. pelliculosa* cells from cultures in fresh water and grown at 400 ppm CO_2_ was 2-fold higher than in cells cultured in sea water and grown at 20,000 ppm CO_2_. PGK activity from *A. formosa* only slightly, but not significantly, increased in cells grown at 20,000 ppm CO_2_ (Figure 5e). In addition, we did not observe redox regulation in PGK as result of DTT treatment in any of the diatoms tested, in contrast to what was previously described in *P. tricornutum* [38].

#### 3.4.3. Pyruvate Kinase (PK)

The pyruvate kinase (PK) activity was measured in all diatoms studied. The PK activity in all diatom cells grown at 20,000 ppm CO_2_ (Figure 6) was much higher compared to the PK activity in cells grown at low CO_2_. The PK activity in *T. pseudonana* showed a 2-fold increase after high CO_2_ treatments (Figure 6a). The PK activity in *P. tricornutum* was almost absent in cells grown at 400 ppm compared to high CO_2_, where activity increased dramatically (Figure 6b). *N. pelliculosa* cultured in seawater and freshwater had similar PK activities when grown at high CO_2_. However, at low CO_2_ PK activity was 5-fold higher in *N. pelliculosa* grown in freshwater medium compared to sea water (Figure 6c,d). *A. formosa*, showed similar results, including a high PK activity at high CO_2_ that was up to 16-fold compared to low CO_2_.

## 4. Discussion

In this work, we showed that the content of storage compounds varied among diatoms and that the response to high CO_2_ is species-specific. In addition, the effect of CO_2_ can be also affected by the culture media used for diatom growth, at least for diatoms that have the capacity to grow indistinctly in seawater or freshwater environments.

In our experimental conditions, we observed that lipid productivity in *T. pseudonana* is doubled in the stationary phase compared to the exponential growth (Figure 2a). However, neither TAG accumulation nor growth were affected by high CO_2_ compared to low (atmospheric) CO_2_ (Figure 1a and Figure 2a). Our data differs from what other authors have shown about the positive effect of CO_2_ in biomass and lipid content in *T. pseudonana* [27,39,40]. One explanation could be the different experimental setup used in all studies, especially the differences in light intensity, photoperiod, and CO_2_ supply. For instance, we kept our cultures at continuous light and photon flux of ~50 µmol m^−2^ s^−1^, while other authors used higher light fluxes and light photoperiod (light:dark) [11,26,27]. In addition, different CO_2_ concentrations up to 20–30% (200,000–300,000 ppm) have been also reported [11], although it has been observed that very high CO_2_ concentrations and high light can have deleterious effect on growth [40,41].

In *P. tricornutum,* we observed an increase in lipid productivity of 1.7-fold in cells grown at 20,000 ppm CO_2_ (Figure 2b); this value is only slightly higher than the 1.5-fold increase reported by Wu et al. in this species grown at 1500 ppm [42].

In the freshwater diatom *A. formosa*, lipid productivity is very low compared to other diatom species (Figure 2e and Table 1). This might be related to a very low growth compared to other species. In fact, although we observed a high increased growth of *A. formosa* grown at high CO_2_, the OD_750_ reached a plateau of about 0.08, compared to 0.8–1.0 in the other diatoms analyzed. There is no data in the literature about lipid accumulation in *A. formosa* grown at high CO_2_, however an increase in both growth and lipid content has been reported in this species grown in bicarbonate (5 mM, compared to 0.15 mM in control) [13]. This difference suggests that *A. formosa* performance highly depends on the nature of the carbon source supplied to the culture medium and the mechanism used by this diatom to uptake carbon from the environment. In addition, it has been shown that *A. formosa* relies more on the use of CO_2_ as a source of carbon than HCO_3_^−^ [43], which might explain the high effect observed on growth in this diatom at high CO_2_ concentration.

The oleaginous diatom *N. pelliculosa* [44,45] accumulated high amounts of lipids (up to 32% of the total cell dry weight; not shown) and has the highest lipid productivity in the stationary phase compared to all the other diatom species studied here (Figure 2c,d and Table 1). However, we observed important differences when this diatom was cultured in seawater or freshwater. In seawater condition, growth slightly increased at high CO_2_, and lipid productivity in the stationary phase was 40% higher than with cells grown at low CO_2_. Interestingly and in contrast to seawater condition, *N. pelliculosa* cultured in freshwater medium had decreased lipid productivity at high CO_2_ compared to low CO_2_ (Figure 2d). Previous work showed that, at high CO_2_, the affinity for CO_2_ (shown as a reduced half-saturation constant) is 7.5-fold higher in *N. pelliculosa* cultured in seawater than in freshwater, although the overall CO_2_-concentrating mechanisms (CCMs) are not different in these two conditions [43]; thus, the higher lipid accumulation observed in cells grown in seawater medium at high CO_2_ might be more related to a change in the carbon conversion/metabolism than to a change in CO_2_ uptake. *N. pelliculosa* is a diatom species with a high tolerance to salinity as it can grow in fresh water and marine media. Salinity is one of the major chemical factors influencing lipid production in microalgae [46], and in diatoms, a change in lipid composition as a result of high salt stress that might be linked to a modification of the permeability and fluidity of the cell membrane has been observed [47]. Moreover, in the marine diatom, *Chaetoceros gracilis*, a moderate low salt (NaCl) and silicon stress can synergistically induce lipid accumulation, although both treatments have detrimental effects on cell growth [10]. Here, we observed that cell growth of *N. pelliculosa* is not affected by salinity and, therefore, it cannot be considered as a stress condition for this diatom species. This suggests that salinity is able to induce a metabolic reprogramming which affects lipid biosynthesis in response to elevated CO_2_ without affecting biomass accumulation. More work is needed to unravel this metabolic innovation in *N. pelliculosa*, and this might be also relevant for other microalgae species showing high tolerance to varying environmental conditions.

Studies on diatoms show that an impairment in one of the main two storage compounds synthesis pathways could shift the balance towards the synthesis of the other one. For instance, in *P. tricornutum* a reduction of vacuolar chrysolaminarin synthesis caused by a mutation of the vacuolar β-1,3-glucan synthase showed increased TAGs accumulation at both N-replete and N-deplete conditions [48]; similar effects were also observed when the same gene was knocked-down in *T. pseudonana* [49]. On the other hand, upregulation of chrysolaminarin synthesis as a consequence of the overexpression of the phosphoglucomutase gene results in a down-regulation of lipid accumulation [50]. Here, we did not find this negative correlation between lipid and chrysolaminarin production as response to CO_2_ treatment (Figure 2 and Figure 3). In fact, the only diatom species where we observed an increase in chrysolaminarin when grown at high CO_2_—*P. tricornutum—*also had increased lipid production (Figure 2b and Figure 3b). The same effect has been observed in other microalgae. In the green alga *Chlamydomonas reinhardtii* both lipids and starch contents (storage carbohydrate in algae and equivalent to chrysolaminarin in diatoms) increased in cultures grown at 5% (50,000 ppm) [51] and at 2% (20,000 ppm) CO_2_ [52], as well as in *Chlorella sorokiniana* grown at 2% CO_2_ [53]. It is possible that in some microalgae CO_2_ treatment can have a synergistic effect on the accumulation of both storage compounds.

Chrysolaminarin content in *P. tricornutum* was reported by some authors to be around 1.5 pg cell^−1^ [54], this is ten times less than the lowest chrysolaminarin content showed in this work in the same species. Similar was the case of chrysolaminarin content in *T. pseudonana*, where we observe a content between 4–8.5 pg cell^−1^, while in other works only 1.5 pg cell^−1^ was found [49]. This might be also related to the differences on the culture conditions used here, as it was discussed above for lipid productivity. Moreover, in contrast to *P. tricornutum*, in which both lipids and chrysolaminarin were increased at high CO_2_ in the stationary phase, in *T. pseudonana* accumulation of chrysolaminarin was decreased (Figure 3a), which might reveal different adaptation mechanisms in these two marine species. Interestingly, *N. pelliculosa* had the same chrysolaminarin content regardless of CO_2_ or when grown in freshwater or seawater culture, in contrast to what was observed for lipid productivity; unfortunately, there is no transcriptomic or proteomic data on this species that could help to understand the modulation of the chrysolaminarin or lipid biosynthetic pathway in different growth conditions. From all diatoms studied, *A. formosa* showed the highest amount of chrysolaminarin content per cell; this in contrast to an extremely low lipid production. Other diatom species, like the marine *Chaetoceros pseudocurvicetus*, also have much higher chrysolaminarin accumulation than lipids [55]. In addition, other freshwater microalgae—like *Chlorella* sp., *Arthrospira* sp., and *Chlorococcum* sp.—also have preference to store carbohydrates (starch) than lipids [53,56,57,58]. This preference for chrysolaminarin accumulation over lipids in *A. formosa* needs to be further investigated.

Regulation and activity of the chloroplast GAPDH was different in all the diatoms studied (Figure 4). Redox regulation of GAPDH mediated by the formation of an inhibitory complex formed in oxidized condition and involving GAPDH, phosphoribulokinase (PRK), and the small intrinsically disordered CP12 that has been well studied in some plants and green algae [59,60,61,62,63]. Here, we showed that in *T. pseudonana* and *P. tricornutum* GAPDH is not regulated by DTT (reduced conditions) alone; however, activity is boosted after incubation with both DTT and NADPH, similar to previous studies in the case of *T. pseudonana*, although a lower activity value was reported [64,65]. In *P. tricornutum,* the formation of the regulatory complex GAPDH/PRK/CP12 has been already discarded [66], and there is no evidence of its formation in *T. pseudonana*. Moreover, it has been shown that the formation of this ternary complex requires the presence of two cysteine residues on PRK, that are absent in diatom PRK [67]. This suggests that regulation of GAPDH in these two diatoms relies on a different mechanism. In contrast, *N. pelliculosa* is highly regulated by reduction, using DTT alone, suggesting that a complex similar to the GAPDH/PRK/CP12 is feasible, but no sequences are available and this needs to be further studied. In *A. formosa*, a regulatory complex involving GAPDH, CP12, and ferredoxin-NADP reductase (FNR) has been observed [68]. In *P. tricornutum* and *N. pelliculosa* cultured in seawater, GAPDH activity increased by 7- and 8-fold, respectively, when cells were grown at high CO_2_ concentrations (Figure 4b,c). This increase in GAPDH activity correlates with the increased lipid productivity observed in these two species. GAPDH activity and lipid accumulation are also shown to increase in *A. formosa* cultured in the presence of plant hormones and high bicarbonate concentrations [13]. This is consistent with GAPDH producing trioses-phosphate that are required building blocks for lipid biosynthesis. In addition, we did not observe any significant increase neither in lipid productivity nor in GAPDH activity in both *A. formosa* and *T. pseudonana* grown at high CO_2_.

The pattern of PGK activity, as that of GAPDH, also correlates with lipid productivity in diatoms (Figure 5). *P. tricornutum* and *N. pelliculosa* in seawater culture showed both higher lipid productivity and higher PGK activity in cells grown at high CO_2_ (Figure 2b,c, and Figure 5b,c). In contrast, in *T. pseudonana* (Figure 2a and Figure 5a) and *A. formosa* (Figure 2e and Figure 5e), PGK activity and lipid productivity were rather similar in cells grown at high or low CO_2_. Wu et al. [42] showed that in *P. tricornutum* grown at high CO_2_ concentrations, PGK mRNA increases 3-fold. Furthermore, increased PGK activity is associated with oil accumulation in sunflower seeds during the synthesis of storage lipids [69]. In agreement with this, we found a positive correlation in *N. pelliculosa* cultured in freshwater media and grown at high CO_2_ concentration, with a decrease in both PGK activity and lipid productivity (Figure 2d and Figure 5d).

PK activity increased in all the diatoms tested. Thus, it is not clear whether there is a correlation in PK activity and lipid productivity. In some higher plants and fungi that PK expression is directly related to lipid biosynthesis and accumulation [70,71,72,73]. In contrast, in mammals, a PK isoform that is expressed in tumor cells (PKM2) does not affect lipid production when the gene that encodes this protein is knocked-out [74]. Moreover, in the green alga *C. reinhardtii* there is no change in the gene expression of PK in cells that accumulate TAGs as response to N-deprivation [75]. On the other hand, a proteome analysis of *P. tricornutum* during lipid accumulation induced by N-deprivation, showed upregulation of three PK isoforms [76], which agrees with our observations for this diatom species, however PK might not be involved in the same way in lipid accumulation in the other diatom species studied here.

To conclude, diatoms are important primary producers within phytoplankton communities [1,77], however, compared to land plants, there are fewer studies that emphasize the effect of CO_2_ in marine and freshwater microalgae and its ecological relevance [78,79,80]. The results of this work show that CO_2_ change is tackled differently by the different diatoms we have studied. Responses at the protein levels (activity and regulation) and storage compounds levels varied between freshwater and seawater diatoms under uniform conditions. Moreover, increasing the knowledge on genomes of diatoms from different aquatic ecosystems is essential to understand their role in the global biogeochemical cycling of carbon, especially when related to modern issues on climate change as result of human activity [81,82,83,84]. This work therefore brings some new elements on CO_2_ effect on some diatoms using short term experiments and reinforces the need to study more diatoms for a better understanding of the different mechanisms (carbohydrate, lipid synthesis) involved in these ecologically important organisms.

## Figures and Tables

**Figure 1 biology-09-00005-f001:**
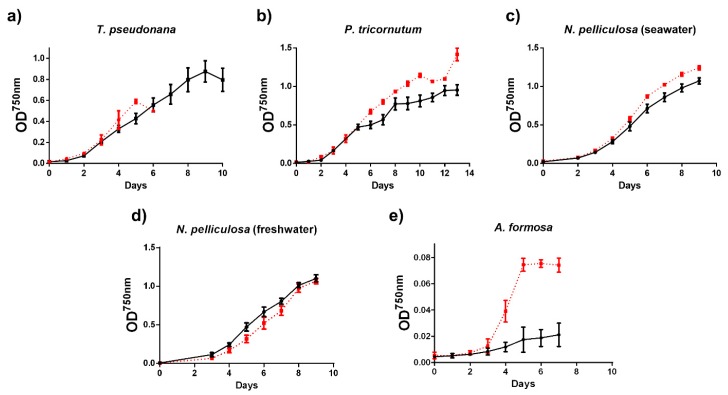
Growth curves of diatoms (a–e) grown at low (400 ppm; black and solid lines) or high (20,000 ppm; red and dotted lines) CO_2_. Each point represents the average (*n* = 3) ± SD.

**Figure 2 biology-09-00005-f002:**
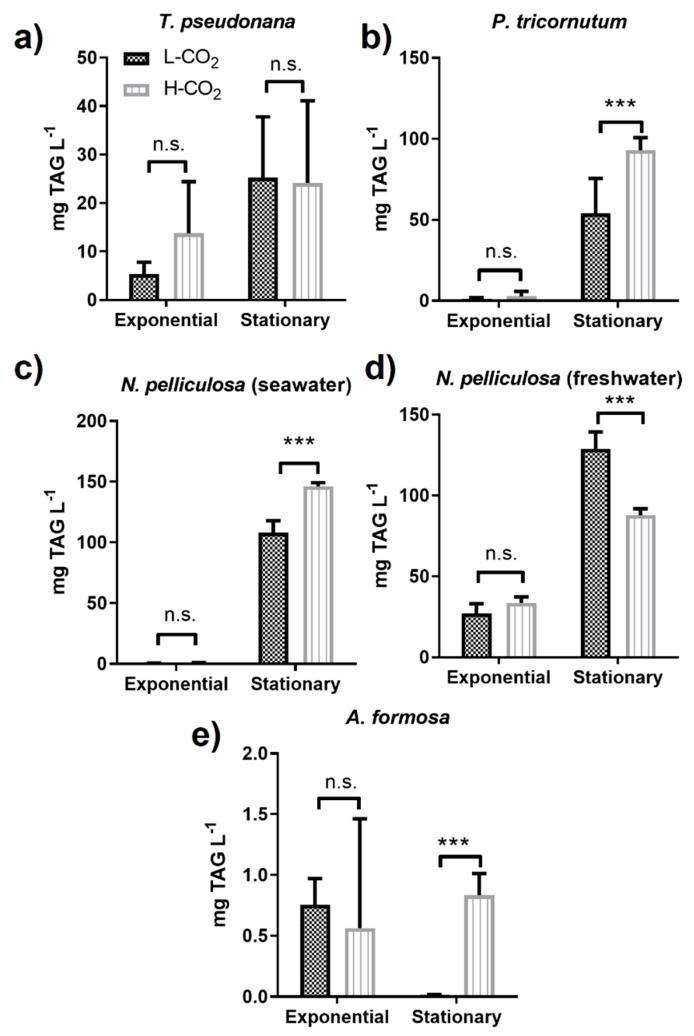
Triacylglycerol (TAG) productivity in diatoms (**a**–**e**) grown at different CO_2_ concentrations. All panels represent low CO_2_ (L-CO_2_; 400 ppm, dark grey bars) and high CO_2_ (H-CO_2_; 20,000 ppm, light grey and dashed bars) as in panel (**a**). Average and error bars (SD) are represented (*n* = 3). *** *p* ≤ 0.005.

**Figure 3 biology-09-00005-f003:**
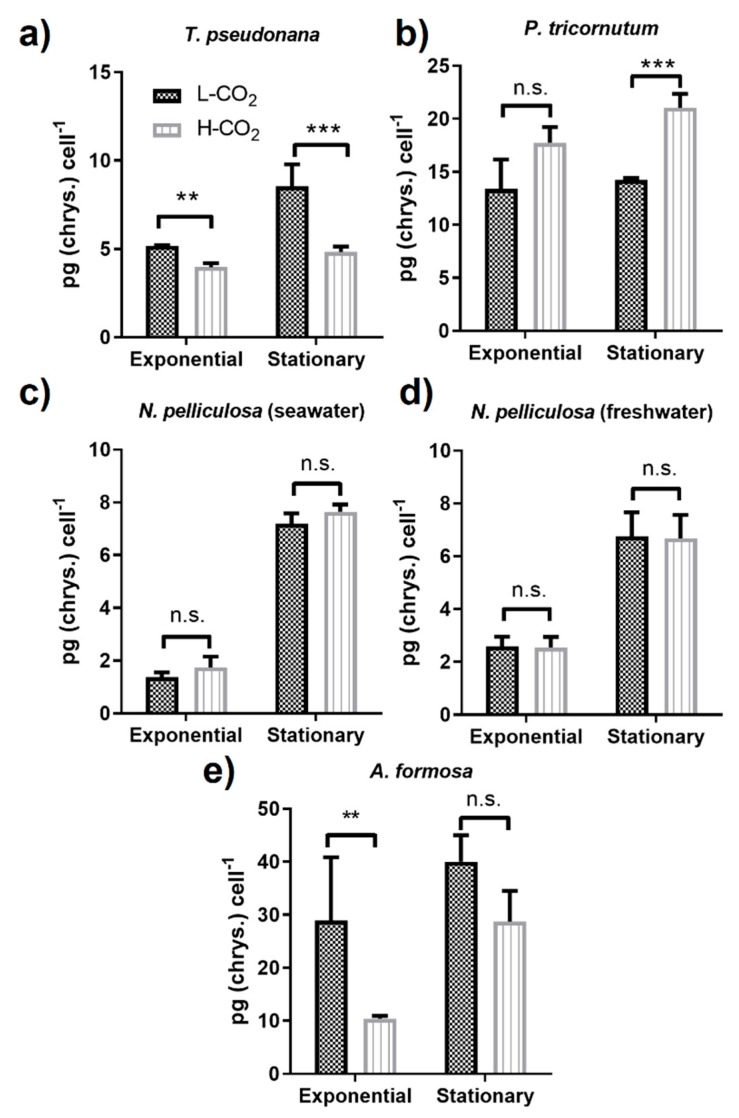
Production of chrysolaminarin in diatoms (**a**–**e**) at different CO_2_ concentrations. All panels represent low CO_2_ (L-CO_2_; 400 ppm, dark grey bars) and high CO_2_ (H-CO_2_; 20,000 ppm, light grey and dashed bars) as in panel (**a**). Average and error bars (SD) are represented (*n* = 3). ** *p* ≤ 0.01, *** *p* ≤ 0.005, and ns, not significant.

**Figure 4 biology-09-00005-f004:**
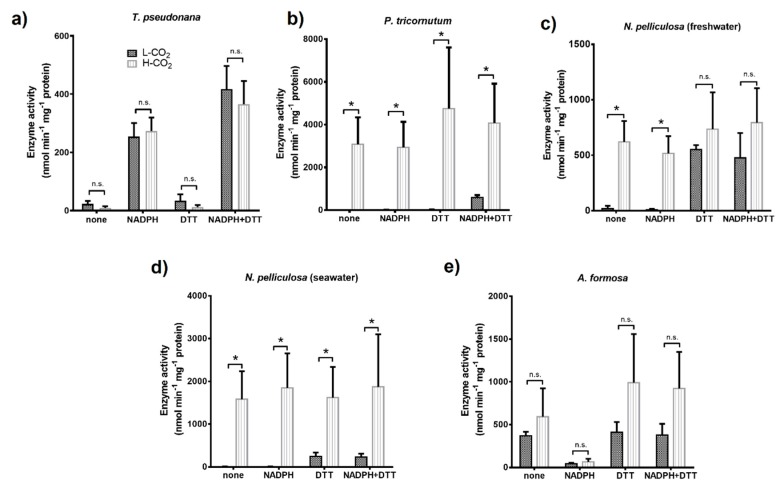
NADPH-dependent activity of chloroplast glyceraldehyde 3-phosphate dehydrogenase (GAPDH) in diatoms (**a**–**e**). All panels show activity measurements of diatom crude extracts from cultures grown at low CO_2_ (L-CO_2_; 400 ppm, dark grey bars) or high CO_2_ (H-CO_2_; 20,000 ppm, light grey and dashed bars), as specified in panel (**a**), and without any treatment (none) or treated with either 0.2 mM NADPH or 5 mM DTT, or both prior to assay. Average and error bars (SD) are represented (n = 3). * *p* ≤ 0.05 and ns, not significant.

**Figure 5 biology-09-00005-f005:**
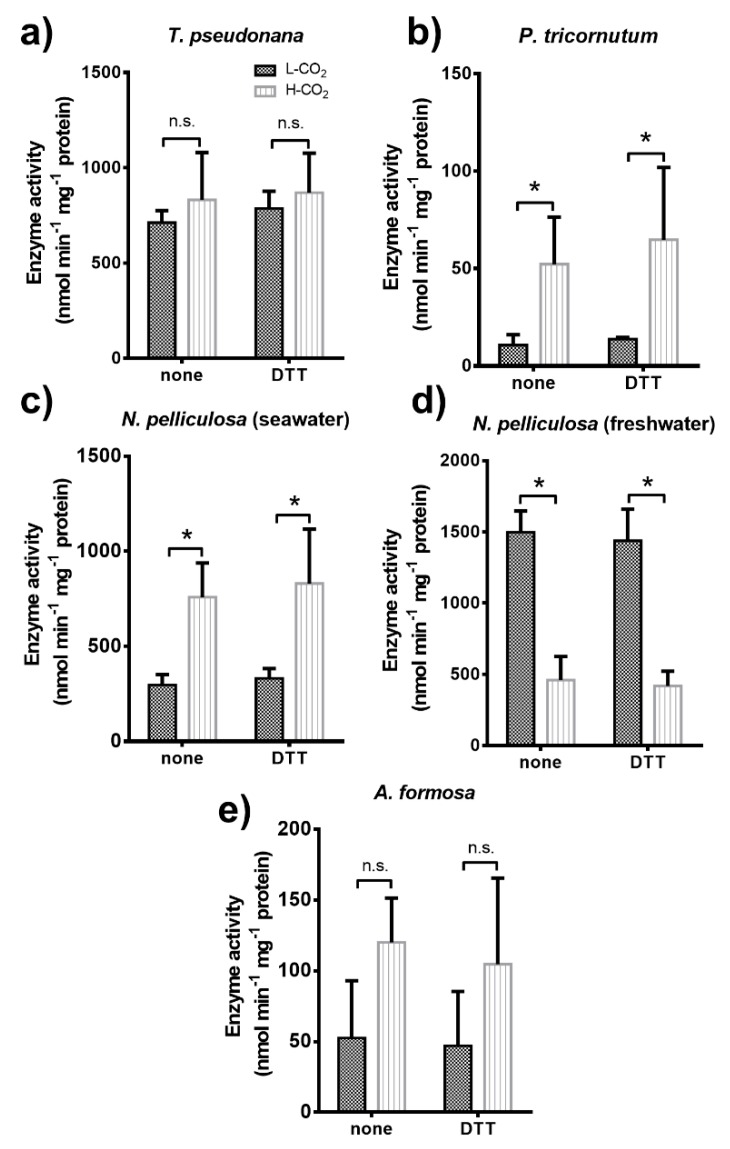
Phosphoglycerate kinase (PGK) activity in diatoms (**a**–**e**). All panels show activity measurements of diatom crude protein extracts from cultures grown at low CO_2_ (L-CO_2_; 400 ppm, dark grey bars) or high CO_2_ (H-CO_2_; 20,000 ppm, light grey and dashed bars), as specified in panel (**a**), and without any treatment (none) or treated with 5 mM DTT prior to assay. Average and error bars (SD) are represented (n = 3). * *p* ≤ 0.05 and ns, not significant.

**Figure 6 biology-09-00005-f006:**
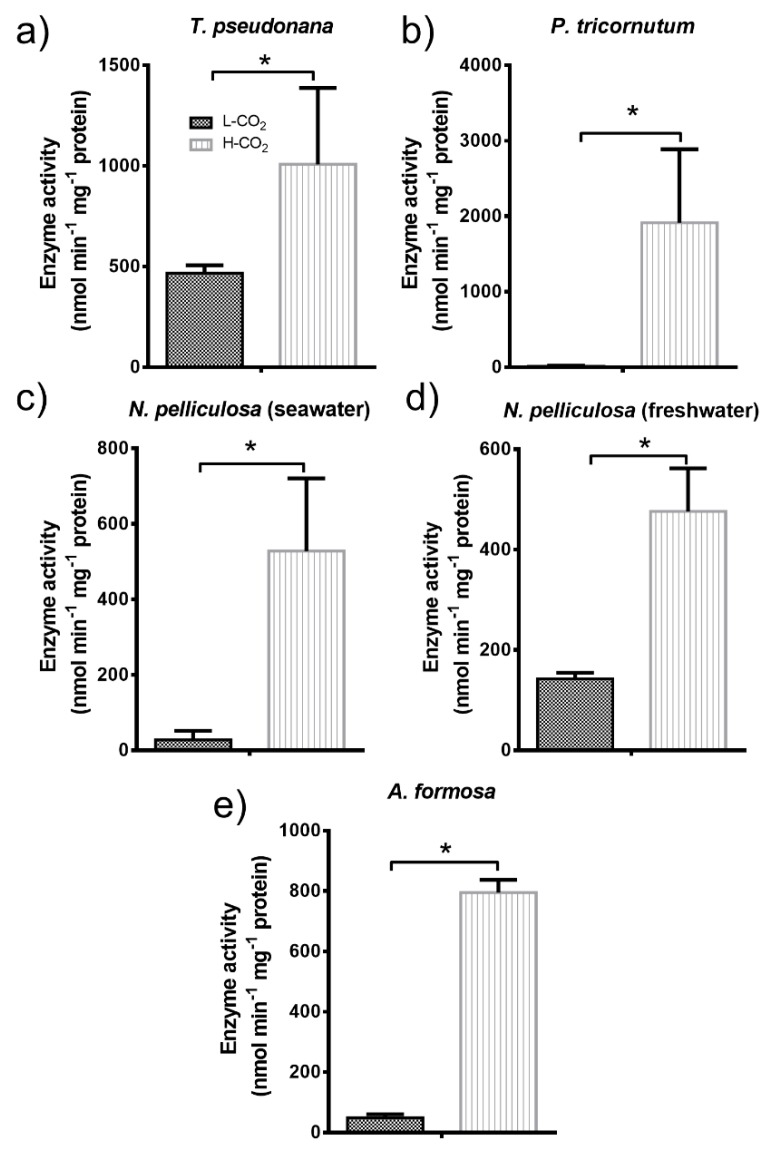
Pyruvate kinase (PK) activity in diatoms (**a**–**e**) grown at low CO_2_ (L-CO_2_; 400 ppm, dark grey bars) or high CO_2_ (H-CO_2_; 20,000 ppm, light grey and dashed bars), as specified in panel (**a**). Average and error bars (SD) are represented (n = 3). * *p* ≤ 0.05.

**Table 1 biology-09-00005-t001:** Biomass and lipid (TAG) productivity and content in diatoms grown at different CO_2_ concentration.

Diatom Species	Growth Phase	CO_2_ Concentration (ppm)	Biomass Productivity (mg L^−1^ day^−1^) *	TAG Productivity (µgTAG L^−1^ day^−1^) *	Percentage of TAGs (DW) *
*T. pseudonana*	Exponential	400	61.7 ± 7.0	11.8 ± 3	2.3 ± 1.0
		20,000	68.6 ± 9.0	45.5 ± 26	5.3 ± 4.2
	Stationary	400	45.3 ± 3.0	13.6 ± 2	5.5 ± 3.3
		20,000	55.8 ± 2.0 ^b^↑	19.0 ± 5	7.2 ± 5.3
*P. tricornutum*	Exponential	400	28.5 ± 4.0	3.36 ± 1	0.67 ± 0.65
		20,000	36.7 ± 6.0	8.47 ± 7	1.2 ± 1.4
	Stationary	400	51.0 ± 4.0	50.2 ± 15	10.1 ± 3.6
		20,000	56.9 ± 1.0	75.7 ± 9	13.5 ± 1.3
*N. pelliculosa (seawater)*	Exponential	400	40.7 ± 4.0	0.03 ± 1	-
		20,000	49.3 ± 3.0 ^a^↑	1.09 ± 1	0.22 ± 0.2
	Stationary	400	48.8 ± 5.0	117.4 ± 12	25.7 ± 3.0
		20,000	58.6 ± 1.0 ^b^↑	158.4 ± 29 ^b^↑	27.0 ± 0.8
*N. pelliculosa (freshwater)*	Exponential	400	42.8 ± 13.0	44.9 ± 10	11.5 ± 5.0
		20,000	47.6 ± 12.0	55.8 ± 6	12.0 ± 1.8
	Stationary	400	45.1 ± 8.0	142.3 ± 12	32.0 ± 5.0
		20,000	38.2 ± 1.0	97.25 ± 5 ^b^↓	25.5 ± 0.7
*A. formosa*	Exponential	400	6.4 ± 1.0	1.43 ± 0.4	2.2 ± 0.3
		20,000	5.6 ± 2.0	1.08 ± 1.7	1.4 ± 2.0
	Stationary	400	2.3 ± 1.0	0.01 ± 0.003	0.06 ± 0.01
		20,000	2.5 ± 4.0	0.40 ± 0.8	1.7 ± 0.3

* All data is shown as average (n = 3) ± SD. ^a^ Significant difference between high and low CO_2_ in diatoms at exponential phase for a given species. ^b^ Significant difference between high and low CO_2_ in diatoms at stationary phase for a given species. ↑ and ↓ indicate increase or decrease, respectively. DW: Dry weight.

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
