# Peer review of "Storage Compound Accumulation in Diatoms as Response to Elevated CO2 Concentration"

_biology, 2019, doi:10.3390/biology9010005_

Round 1
Reviewer 1 Report
Over the past 2 decades, extensive laboratory and field research has been directed towards understanding the effects of atmospheric CO2 variations on the productivity and species composition of terrestrial plant communities. By comparison, very few studies have examined the CO2 responses of marine and freshwater phytoplankton despite the established importance of these organisms in the global C cycle. Here, the authors report the results of shor-term experiments demonstrating a CO2 effect on the species phytoplankton species.
In the section on statistical methods one could, and I think it is necessary, explain in more detail the experimental design followed. Sample population size? How many replicates, if any, have been collected? This might be relevant because of the problem with getting suitable significance levels, neither permutation tests nor rank tests may be suitable, and at the smallest sizes, actually a t-test may be the best option (there’s some possibility of slightly robustifying it). However, there’s a good argument for using higher type I error rates with small samples (otherwise you’re letting type II error rates inflate while holding type I error rates constant).
On the other hand, from an editorial point of view, it might be convenient to maintain the design of the graphics in all figures, for the sake of clarity.
Finally, despite the important effects that this increase in carbon can have on some important plankton organisms such as diatoms, an important primary producer in the ocean.However they do not discuss the potential ecological, biogeochemical and paleoecological implications of their findings. I would suggest the authors to bring up their thoughts on the potential for plankton organisms to adapt to increasing CO2 and broader implications for ecosystems.
Reviewer 2 Report
The manuscript, “Storage compounds accumulation in diatoms as response to CO2 concentration”, shows the results of lipid production, carbohydrate production, and enzyme activity in a variety of diatom species in response to elevated CO2. I found that, overall, the manuscript was relatively clear, but could use some light English editing (see minor comments). Certainly, the topic of microalgal response to increasing CO2 is important to both basic and applied scientists. However, the manuscript currently doesn’t relate the significance of the findings in the abstract or in the conclusion. The manuscript could increase its impact by better communicating how their findings are broadly important in a rapidly changing world. The authors’ experimental methods are appropriate and well described. However, there needs to be further explanation of how the concentrations of chrysolaminarin is calculated (see major comments). I found that the authors often did not support their discussion conclusions with direct evidence from their results (a lack of citing the figures in the discussion – specifics below). Overall, I found the manuscript to be scientifically sound, and with some revisions I believe it will be of interest to algal and climate researchers.
MAJOR COMMENTS
Abstract: From line 19-23 the results reported could use additional specificity (i.e., how did they differ, how did enzyme activity change, specifically what impact did elevated CO2 have). In addition, the abstract (and conclusion) needs to include the significance of these findings.
Line 14 – Accumulation of what? I am assuming lipids and carbohydrates.
Results: 3.3 and figure 3 – it appears that the extraction was done on bulk culture material, but it is reported in pg per cell. From the methods, I am not sure how they got this measurement per cell when no cell counts were done before extraction.
Discussion:
Throughout the manuscript, but especially in Line 277: In discussing differing types of culture media as a driver of lipid/carbohydrate production, I believe it would make more sense to discuss this in terms of salinity content (unless there are other properties in the culture medium that you think might be driving these differences).
Overall, it was hard for me to verify the validity of the manuscript’s arguments because often statements in the discussion didn’t include evidence from the figure(s) and/or table. Often, I would go to the effort of finding the specific figures that matched the manuscript discussion’s claims, but the readers shouldn’t have to work so hard.
Examples:
Line 280: provide the figure(s) or table that show this.
Line 281: provide the figure(s) or table that show this.
Line 290: provide the figure(s) or table that show this.
Line 305: provide the figure(s) or table that show this.
Line 309: provide the figure(s) or table that show this.
Line 323: provide the figure(s) or table that show this.
Line 372: provide the figure(s) or table that show this.
Line 373: provide the figure(s) or table that show this.
Line 374: provide the figure(s) or table that show this.
Line 279: provide the figure(s) or table that show this.
Figures: On figure 4 the dark color is labeled as high CO2 and the light color is labeled as low CO2. This is in reverse order of all of the other figures. I would suggest using the same color and order for low and high CO2. Also make sure that figure 4 has not been accidentally swapped.
MINOR COMMENTS
Title: Maybe consider “Storage compound accumulations in diatoms in response to elevated CO2 concentrations”
Line 104-105: Consider rewording for clarity.
Line 62-66: This sentence is quite long and makes it hard to understand.
The manuscript needs a light editing of English language usage. Examples include:
Line 29: add ‘are’ after lipids.
Line 30: ‘amount’ should be plural
Line 40: add ‘the’ before ‘expenses’ and make ‘expenses’ singular
Line 45: it appears that the word ‘increase’ belongs before ‘of lipid accumulation’
Line 62: replace ‘since’ with ‘for’
Line 69: replace ‘on’ with ‘in’
Line 177: I believe you mean ‘present’ instead of ‘absent’
Line 196: replace ‘ranging’ with ‘ranged’
Line 263: I believe you mean ‘present’ instead of ‘absent’
Line 327: Missing end parentheses after diatoms.
